# A Vismodegib Experience in Elderly Patients with Basal Cell Carcinoma: Case Reports and Review of the Literature

**DOI:** 10.3390/ijms21228596

**Published:** 2020-11-14

**Authors:** Anna Passarelli, Giovanna Galdo, Michele Aieta, Tommaso Fabrizio, Antonio Villonio, Raffaele Conca

**Affiliations:** 1Department of Onco-Hematology, Unit of Medical Oncology, Centro di Riferimento Oncologico della Basilicata (IRCCS-CROB), 85028 Rionero in Vulture, Italy; michele.aieta@crob.it (M.A.); raffaele.conca@crob.it (R.C.); 2Department of Onco-Hematology, Oncologic Dermatology Unit, Centro di Riferimento Oncologico della Basilicata (IRCCS-CROB), IRCCS-CROB, 85028 Rionero in Vulture, Italy; giovanna.galdo@crob.it; 3Division of Plastic Surgery, Centro di Riferimento Oncologico della Basilicata (IRCCS-CROB), IRCCS-CROB, 85028 Rionero in Vulture, Italy; tommaso.fabrizio@crob.it; 4Radiology Department, Centro di Riferimento Oncologico della Basilicata (IRCCS-CROB), IRCCS-CROB, 85028 Rionero in Vulture, Italy; antonio.villonio@crob.it

**Keywords:** basal cell carcinoma, non-melanoma skin cancers, Hedgehog pathway, elderly patients, vismodegib, sonidegib

## Abstract

Cutaneous basal cell carcinoma (BCC) is the most common type of human tumor, and its incidence rate is increasing worldwide. Up until a few years ago, therapeutic options have been limited for patients with advanced BCC (including metastatic and locally-advanced BCC). Over the last few years, promising systemic therapies have been investigated for the treatment of advanced BCC. In particular, the Hedgehog signaling inhibition has shown remarkable results for this population. Hedgehog inhibitors, represented by vismodegib and sonidegib, have been approved by the Food and Drug Administration and the European Medicines Agency for the treatment of both locally advanced and metastatic BCC, with, generally, a good safety profile. Notwithstanding the late onset of BCC in the global population, associated with life expectancy increase, only a few clinical trials have evaluated the efficacy and safety profile of Hedgehog inhibitors in this complex and neglected population. Herein, we review the major mechanisms implicated in the pathogenesis of BCC focusing on the Hedgehog signaling pathway and its therapeutic role in the elderly population. Finally, we report two case reports of BCC elderly patients in order to demonstrate both efficacy and safety of the Hedgehog inhibitors.

## 1. Introduction

Non-melanoma skin cancers (NMSCs) include basal cell carcinoma (BCC) and squamous cell carcinoma (SCC), representing the most common cancers in the Caucasian population [1].

BCC, first described in 1827 [2], is the most commonly diagnosed skin cancer worldwide [3]; thus, comprising approximately 80% of NMSCs. The incidence of BCC is increasing worldwide, by approximately 1% annually [4]. The increase in incidence of BCC could be due to changes in the environment or lifestyle risk factors. The most well-known risk factor is represented by solar ultraviolet (UV) exposure, especially early in life, or as a result of intermittent exposure.

For most BCCs, including small, well-defined tumors or intermediate-sized, low-risk tumors in low-risk areas, primary treatment options usually include surgical excision, Mohs micrographic surgery, cryotherapy, and radiation therapy. Thus, the multidisciplinary approach is critical in the management of patients suffering from BCC.

Although BCC could be effectively counteracted by radical surgical excision, sometimes it can acquire aggressive hallmarks, as well as recurrence, local tissue destruction, and (infrequently) distance dissemination. In a subset of cases, BCC can become excessively invasive or destructive; these are known as advanced BCC, and are defined as BCC in which current treatment modalities are contraindicated. Patients affected by advanced BCC, including locally-advanced BCC (laBCC) or metastatic BCC (mBCC), typically had, up until a few years ago, limited therapeutic options.

In particular, advanced BCC is a common cutaneous malignancy, mostly occurring in elderly populations [5]. To this regard, factors, including social isolation and multiple comorbidities, leave the elderly population at high risk of neglecting potentially malignant skin cancers, causing them to progress to advanced diseases [6]. In this context, several factors should be considered in order to plan BCC treatment, including tumor location, size, and nature of the lesion [7]. Other clinical factors, such as symptoms, age, and performance status of the patient, as well as the cost of therapy, are also critical. Moreover, the treatment should ideally include removal of the whole tumor, with preservation of healthy tissue, function, and cosmetic appearance.

In the last few years, promising therapeutic strategies have emerged for the treatment of advanced BCC. In this regard, aberrant signaling of the Hh pathway proved critical to the carcinogenesis and progression processes of BCC, and two different Hh signaling inhibitors, vismodegib and sonidegib, have been approved for advanced BCC treatment [8]. In the pivotal study, vismodegib demonstrated response rates of 43% in patients with laBCC, associated with a median duration of response of approximately six months.

Unfortunately, few clinical data regarding both safety and efficacy of Hh inhibitors are currently available for the elderly population.

Herein, we review the most relevant findings in the management of BCC, focusing on pathogenic and clinical features, as well as novel therapeutic strategies in the elderly BCC population. Finally, we describe two case reports of elderly patients affected by laBCC and treated with an Hh inhibitor, which has clinical benefits and an acceptable safety profile.

## 2. Local Treatment Options for BCC

Current BCC clinical practice guidelines focus on the curative intent of removing as much malignancy as possible. Unfortunately, there are no clear recommendations in clinical practices for a tailored approach for elderly patients who have special needs and priorities.

Several treatment options, including surgical excision, electrodessication and curettage, cryosurgery, imiquimod, photodynamic therapy, 5-fluorouracil, radiation therapy, Hh inhibitors, combination therapy, and observation may be considered in the BCC management. 

Given the wide range of therapeutic options, the choice of the best treatment should be tailored to meet patient care goals based on their life expectancy.

### 2.1. Surgical Excision

Surgery with negative margins is the standard treatment for localized BCC. It has been reported that larger surgical margins, as well as the negative margin on histopathological examination, ensure better outcomes [9]. Indeed, a systematic analysis has reported that recurrence rates for 5-, 4-, 3-, and 2-mm surgical margins are 0.39, 1.62, 2.56, and 3.96 percent, respectively [10]. In addition, cumulative recurrence rate for primary BCC after surgical excisions also seems to depend on the anatomic tumor site [11]. Nevertheless, total excision has been reported to have a 5-year cure rate for BCC, as high as 98% [12]. 

In the surgical management of BCC in elderly populations, several factors should be taken into account. For example, wound healing after surgical procedures in geriatric patients is not as smooth as in the younger population. Indeed, it has been reported that elderly patients (>65 years) show a significant delay in the epithelialization process after a split-thickness wound, leading to increased infection risk [13,14]. Epithelialization in the elderly can be complicated by common multiple comorbidities, such as vascular disease and diabetes mellitus [15]. Furthermore, the evaluation of the elderly patient’s compliance is strongly decisive in the choice of the best procedure. 

Mohs micrographic surgery (MMS), a complete surgical excision with examination of margins in horizontal sections, is the preferred surgery technique for high risk BCC, because it allows the intraoperative analysis of 100% of surgical margins (differently from standard vertical margins examination). High-risk tumors include BCC on the central aspect of the face, as well as large, recurrent, or aggressive lesions in cosmetically or functionally important areas. The cure rate for MMS is estimated at 99% [12]. MMS is associated with a recurrence rate 1% and 5.6% for primary and recurrent BCC, respectively [16,17]. This recurrence rate is lower than any form of alternative local treatment.

Interestingly, both Mohs and standard surgery are well tolerated in elderly patients [18,19]. Several studies analyzed the surgical outcome of BCC in elderly patients, reporting that the BCC extension (>1 cm^2^), the aggressive histology (morpheaform and micronodular), and an age over 80 years are strong predictors for two or more MMS procedures to achieve complete excision [20,21]. However, advanced age would not seem to affect the recurrence rate and the survival rate after surgery [22,23,24].

### 2.2. Radiation Therapy

Any type of radiation therapy, including superficial, conventional, or brachytherapy is an alternative to surgery for BCC that is not able to be eradicated with surgery, or for elderly patients with severe comorbidities who refuse it. Some studies suggest that radiotherapy has a 5-year cure rate for BCC, of 93% to 96%, which is comparable to the surgical excision in terms of cure rate [12]. However, the efficacy of radiotherapy is lower than surgical approach in terms of local relapse [25]. Adjuvant radiotherapy should be considered after primary resection in case of cartilage invasion or both perineural and bone infiltration.

### 2.3. Destructive Therapies

Destructive therapies with curettage and electrocautery (electrodessication), cryotherapy, cryosurgery, and laser treatment are therapeutic options for small and low-risk non-facial BCC.

Curettage and electrodesiccation: curettage and electrodesiccation are recommended treatments of choice for low-risk primary BCC. The therapeutic procedure is based on intradermally anesthetized skin, followed by curettage, alternating with apply light electrodesiccation. The procedure is aimed at removing all of the soft and friable tumor material [26]. Although it has been associated with cure rates as high as 97% to 98.8%, the efficacy of curettage and electrodessication is highly operator-dependent [27,28]. 

Cryotherapy and cryosurgery: cryotherapy is the destruction of tissue by the direct application of a cryogenic agent, such as liquid nitrogen. Therefore, this approach is useful to treat a wide variety of skin conditions, including small, multiple, and low-risk BCC [29,30]. 

In cryosurgery, tissue destruction is caused by freezing leading to sudden loss of heat and subsequent vascular stasis and cell death, thus, representing an appropriate and easily available therapeutic modality with hardly any contraindications [31]. Cryosurgery is cost-effective, requiring minimal anesthesia. Finally, cryosurgery has cure rates as high as 99% [12]. 

Laser treatment: several types of laser, using both selective and ablative methods, are utilized in oncology [32]. Carbon dioxide (CO2) and erbium yttrium aluminum garnet (Er: YAG) lasers remove tumors by the vaporization of water element [33]. A pulsed dye laser, on the other, selects tumor vasculature, so it destroys the tumor’s blood supply [34]. Laser ablation with CO2 may be able to complete the remission of superficial BCC similarly to what happens for cryotherapy [35]. The response to the treatment, in the absence of a histological examination, evaluated under a confocal microscope, is able to demonstrate the absence of the tumor in the residual tissue [36]. However, studies are still ongoing on the effectiveness of combined laser treatment on BCC.

### 2.4. Topical and Intralesional Therapies

Imiquimod therapy: imiquimod is an immune response modifier used for the treatment of small BCC (sBCC) [37]. The mechanism of action of the drug is not yet fully elucidated. Imiquimod, acting as an agonist of the toll-like receptor 7 (TLR7), enhances dendritic cell survival and promotes the activation of tumor-specific T-lymphocytes [38]; thus, inducing secretion of several cytokines, such as interferon-α, interleukin-6, and tumor necrosis factor-α. The cream is applied 5 days/week for six weeks. Imiquimod represents a useful treatment for low risk, single or multiple sBCC, with a maximum diameter of 2 cm. It has been reported that imiquimod has a cure rate of 83% [12]. Therefore, imiquimod could represent a valid alternative in the treatment of elderly patients with major comorbidities and poor compliance. 

Photodynamic therapy: PDT with 5-aminolevulinic acid (ALA) or its methyl ester (methyl-5-amino-4-oxopentanoate, MAL) can be effective for small and superficial BCC, with a thickness not exceeding 2 mm, if the surgical intervention is not indicated for lack of radicality or for the patient’s health conditions (age and comorbidity, drugs) [39]. PDT is a therapy for the non-surgical treatment of actinic keratosis, superficial BCC, and recently, Bowen’s disease. The therapeutic rationale exploits a photodynamic reaction on the tumor cells for the interaction of light with a photosensitizing substance, which, when activated, releases species reactive oxygen, capable of destroying the cell in which they formed. In elderly patients or with contraindications to surgery, combined therapies could represent the gold standard for BCC treatment. Nevertheless, studies are contradictory regarding the effective role of PDT in the elderly population. To this regard, it is still debated whether the elderly population respond differently to treatment with PDT. Moreover, acute post-procedure hypertension is reported as a potential side-effect of PDT, most prevalent in elderly populations, especially in patients with hypertension in their medical history. 

Finally, 5-Fluorouracil: 5-FU, with a 5% cream formulation is advised in the treatment of superficial BCC with the following dosage schedule: 2 daily applications for 2–4 weeks. High response rates have been reported in the literature, in particular for sBCC. Indeed, it has been reported that 5-FU has a 5-year cure rate of 80% [12]. Regarding the real effectiveness of this topical therapy on sBCC, a direct comparison between 5-FU, imiquimod and MAL-PDT shows that imiquimod has a higher probability of success than 5-FU, except for sBCC on lower extremities in older patients [40]. In addition, the application of 5-FU could be difficult for elderly patients with lesions on hard-to-reach locations. 

## 3. Role of Hedgehog Signaling Pathway in BCC 

The skin contains several types of stem cells that participate in the homeostasis of the various epidermal tissues, including hair follicles, sebaceous glands, and interfollicular epidermidis. 

During embryogenesis, the Hh signaling plays a critical role for differentiation and cell proliferation. Indeed, this signaling appears involved in the maintenance of certain tissues and somatic stem cells in adults, including the repair of the skin [41]. To this regard, the Hh pathway is predominantly dormant in the adult organism, but it may be activated during wound healing [42]. 

Interestingly, evidence suggest that BCC arises from basal keratinocytes of the interfollicular epidermis or the hair follicles [43,44]. Abnormal activation of Hh signaling cascading through various mechanisms has been reported in 95% of sporadic BCC [45] and in several human cancers [46]. 

Specifically, Hh signaling in the mammal involves different component, such as three ligands (Sonic hedgehog, Shh; Indian hedgehog, Ihh; Desert hedgehog, Dhh), two different transmembrane receptors (PTCH1, PTCH2), a key signal transducer smoothened, namely smoothened (SMO) and three transcription factors (Gli1, Gli2, Gli3) (see Figure 1).

Simplifying in the normal physiological state, in which the Hh ligands are not present, SMO function is inhibited by another transmembrane protein Patched (PTCH1, PTCH2). Indeed, PTCH is a 12-pass transmembrane receptor protein, which acts such as a tumor suppressor and constitutionally suppresses the Hh cascade. Upon binding of an active Hh ligand, this inhibitory effect is lifted, allowing SMO to signal downstream, eventually leading to active transcription of the glioma-associated oncogene (Gli) transcription molecules through binding the specific sequences located in the promoter region of the target genes [47,48].

The Hh signaling cascade is aberrantly activated in human tumors due to several genetic alterations. To this regard, the genomic characterization of BCC has identified several mutations involved in the regulation of the Hh signaling that may be considered critical for the development or progression of BCC. 

The vast majority of sporadic BCCs are driven by inactivating mutations and loss of heterozygosity in PTCH1. Indeed, several studies show that 73% of BCCs exhibit loss of function of the PTCH1 gene, while about 10% have activating mutations of SMO, thus, inducing an overexpression of Gli, which promotes cell division and the tumorigenesis process. Indeed, genomic mutations in the suppressor PTCH1 gene are involved in the growth of the majority of sporadic BCC, and those that develop it due to the Gorlin syndrome. Gorlin or nevoid basal cell carcinoma syndrome (NBCC) is a rare autosomal dominant disease characterized by the development of multiple BCCs from an early age. While PTCH1 is the primary Shh receptor, PTCH2 has a minor compensatory function in Shh signal transduction, although its function is yet to be fully understood. Interestingly, it has been suggested that PTCH2 variants can also cause NBCCS, albeit with a milder phenotype [49].

Further, p53 mutations are found in sporadic and inherited BCC and can be accompanied by PTCH1 alterations [50]. Furthermore, the translocation of Gli involves the disassociation of the complex from its inhibitor, suppressor of fused (SUFU). In sporadic BCC, the loss of function mutation in *SUFU* has also been described, which leads to increased Gli transcription [51].

The development of therapeutics for the canonical Shh signaling pathway has primarily focused on targeting SMO and Gli. While most efforts have been devoted towards pharmacologically targeting SMO, Gli-targeted approaches are under development. To date, two SMO inhibitors, vismodegib and sonidegib, has been approved for advanced BCC treatment. However, these inhibitors work and are effective only if the mutation is upstream of SMO, thus, these are ineffective in case of SMO or SUFU mutations. Finally, primary resistance to SMO inhibitors is possibly due to variant genes downstream of SMO. 

In the last few years, significant progress in the understanding of the Hh signaling in the BCC pathogenesis and the discovery of Hh inhibitors has led to the introduction of a new treatment strategy that may significantly improve clinical outcomes in advanced BCC. 

## 4. Hedgehog Pathway Inhibitors

### 4.1. Vismodegib

Vismodegib (Erivedge^®^) is the first in class, oral small molecule inhibitor of the Hedgehog pathway. In the USA, and similarly in the EU, vismodegib is a drug indicated for the treatment of adults affected by mBCC and laBCC, who have developed recurrence following surgery, or who are not candidates for radiotherapy. The recommended dose of vismodegib is one capsule (150 mg), with or without food, once daily, until disease progression or unacceptable toxicity occurs.

The role of vismodegib, in terms of both efficacy and safety in the treatment of advanced BCC, has been evaluated in several prospective clinical trials, including two international phase II trials: Efficacy and Safety of the Hedgehog Pathway Inhibitor Vismodegib in Patients with Advanced Basal Cell Carcinoma (ERIVANCE BCC), and STEVIE, a U.S.-based expanded access study (EAS) and a U.S.-based disease registry study (RegiSONIC).

Efficacy and Safety of the Hedgehog Pathway Inhibitor Vismodegib in Patients with Advanced Basal Cell Carcinoma (ERIVANCE BCC) is the follow-up multicenter, phase IIb trial that evaluated vismodegib therapy in two different cohorts, unresectable laBCC and mBCC, with objective response rate (ORR) by central review as the primary endpoint [8].

In the primary analysis of ERIVANCE, the concordance between assessment by central review and investigator review was 60%. Specifically, ORR by central review was 42.9% (95% CI 30.5–56.0), with 20.6% complete response (CR) and 22.2% partial response (PR), whereas ORR by investigator review was 60.3% (95% CI 47.2–71.7), with 31.7% CR and 28.6% PR [52]. Final analysis at 39 months reported ORR by investigator review of 60% (95% CI 47–72) and 49% (95% CI 31–66) in laBCC and mBCC, respectively [53] (see Table 1). The most common adverse events (AE)s for the entire duration of the study included muscle spasms (71%), alopecia (66%), and dysgeusia (56%). In addition, 21% and 28% of patients discontinued the study due to AEs and disease progression, respectively [53].

The STEVIE (NCT01367665), an open-label multicenter trial, which evaluated vismodegib 150 mg once daily in 1215 patients with laBCC or mBCC for a median treatment duration of 86 months (range 0–44), demonstrated similar safety results to ERIVANCE study [54] (see Table 2). The primary analysis of STEVIE demonstrated that vismodegib is tolerable in clinical practice and that long-term exposure is not associated with worsening severity or frequency of AEs. To date, STEVIE is the largest trial in advanced BCC.

Interestingly, the phase II MIKIE trial evaluating the safety and activity of two long-term intermittent vismodegib dosing regimens in patients with multiple BCC, and including those with basal-cell nevus (Gorlin) syndrome, showed good activity in long-term regimens. Therefore, these data support the feasibility of both intermittent dosing schedules of vismodegib [55].

In consideration of its peculiar mechanism of action, unfortunately, vismodegib exerted embryolethal and teratogenic effects in a pregnant rat model [56]. Therefore, the use of vismodegib is absolutely contraindicated during pregnancy.

Finally, regarding the vismodegib retreatment in the setting of progression in advanced BCC, Alfieri and colleagues conducted a retrospective study on six advanced BCC patients enrolled in the STEVIE trial who discontinued vismodegib due to progression disease, and were then retreated with the same drug. All patients underwent intercurrent therapies, such as radiotherapy, chemotherapy, and surgery. Disease control rate (CR, PR, and stable disease (SD)) was achieved in 80% of patients following the second vismodegib course, showing that vismodegib rechallenge is feasible and potentially active in advanced BCC patients who previously discontinued the drug due to disease progression [57].

### 4.2. Sonidegib

Sonidegib (Odomzo^®^) is a selective SMO inhibitor that received FDA approval for the treatment of laBCC that recurred after surgery or radiation, or that cannot be treated with surgery or radiation. Thus, sonidegib is now an alternative treatment option to vismodegib.

Sonidegib is a powerful small-molecule antagonist that binds in the same drug-binding pocket of SMO [58]. This SMO inhibitor has peculiar features, such as a high tissue penetration, the ability to cross the blood–brain barrier, good oral bioavailability, and a long half-life of 29.6 days. In contrast, vismodegib has a half-life of 4–12 days. The recommended dose of sonidegib is 200 mg/day [59].

The approval of sonidegib is based on results of the pivotal phase II Basal Cell Carcinoma Outcomes with LDE225 (sonidegib) Treatment (BOLT) trial, which evaluated both efficacy and safety of sonidegib in patients affected by laBCC and mBCC. Specifically, at the 6-month analysis, treatment with sonidegib 200 mg daily demonstrated ORRs by central review of 43% and 15% in patients with laBCC and mBCC, respectively [59]. Subsequent analyses at 12, 18, and 30 months demonstrated persistent efficacy of sonidegib, with ORRs of 58% in laBCC and 8% in mBCC [60]. At 30-months, the most frequently reported AEs of any grade were muscle spasms (54.4%), alopecia (49.4%), and dysgeusia (44.3%). The most recurrent grade ≥3 AEs were increased creatine kinase (6.3%), weight loss (5.1%), and muscle spasms (2.5%).

Recently, the final 42-month analysis of BOLT trial has been reported. This analysis represents the longest follow-up available for an Hh pathway inhibitor. Clinically relevant efficacy results were sustained from prior analyses, with ORR of the approved 200-mg daily dose of 56% and 8% in laBCC and in mBCC, respectively. ORR were higher for patients with laBCC compared with mBCC, potentially due to the aggressiveness of metastatic disease and the high tumor burden. Regarding the safety profile, reported AEs were consistent with the known safety profile of sonidegib, with no new or late-onset safety concerns emerging at 42 months. Therefore, these results confirmed sonidegib as a viable long-term treatment alternative for patients with advanced BCC [60].

Interestingly, there are no clinical trials directly comparing vismodegib with sonidegib. Despite this, studies regarding the pharmacokinetic profiles of sonidegib and vismodegib have shown several differences. The comparisons between the several registration trials were conducted. To this regard, BOLT and ERIVANCE had similar patients, demographic characteristics, and the same primary endpoint (ORR), but there are many differences between these clinical trials.

In particular, in the laBCC patients group the CR rate treated with sonidegib in the BOLT study is comparable to that of vismodegib in the ERIVANCE study [61]. However, in an industry-sponsored analysis comparing the clinical activity of vismodegib (ERIVANCE) and sonidegib (BOLT), sonidegib patients had a higher ORR, longer median progression-free survival (PFS), and longer median duration of response (DOR) [62]. While for patients with mBCC, the ORR for sonidegib was significantly lower than that for vismodegib [63].

Moreover, in the absence of direct head-to-head evidence, these drugs are comparable in terms of clinical outcomes in patients with BCC. Indeed, patients who show both innate and acquired resistance to vismodegib may also show resistance to sonidegib.

In conclusion, Hh inhibitors act by inhibiting proliferation of BCC. In addition, it has been reported that Hh pathway inhibition may also induce the recruitment in tumor microenvironment (TME) of cytotoxic T-cells and the upregulation of major histocompatibility (MHC) class I in BCC cells. This dynamic modulation of TME suggests the potential synergism of Hh inhibitors when administered with immunotherapy such as immune checkpoint inhibitors [64]. Interestingly, the use of anti-PD-1 mAb (cemiplimab^®^) shows promising results for patients with advanced BCC who had progressed on or were intolerant to prior Hh inhibitor therapy.

## 5. Safety and Efficacy of Vismodegib in Advanced BCC Elderly Patients

BCC is the most commonly diagnosed type of cutaneous cancer worldwide. With an increasing elderly population, it can be derived that advanced BCC will become a more prevalent condition, given also its association with cumulative sun exposure. Unfortunately, elderly patients with advanced BCC are underrepresented in cancer clinical trials and often excluded; thus, it is difficult to have evidence-based recommendations in the clinical practice. The elderly population suffering from advanced BCC is often considered at greater risk of AEs due to the concomitant comorbidities and age-related impairment of organ function.

Therefore, the scientific knowledge and evidence currently available to support the use of Hh inhibitors in the elderly population is greatly limited. However, nonsurgical treatment options also require special consideration when treating the elderly population.

To this regard, a single retrospective analysis was conducted by Chang and colleagues [65] in order to evaluate both the safety and efficacy of vismodegib treatment in patients aged >65 years with laBCC or mBCC enrolled in ERIVANCE BCC (47 pts) and EAS (53 pts), compared with younger patients (<65 years). This analysis demonstrated a similar clinical efficacy in all patient cohorts (see Table 3). Moreover, the safety profile of Hh inhibitor in elderly patients seemed to be similar to that found in younger patients. Specifically, the most recurrent AEs in elderly patients versus (vs.) <65 years in both clinical trials were muscle spasms (64% vs. 72%), dysgeusia (70% vs. 73%), and alopecia (55% vs. 61%).

Grade 3–5 AEs in patients aged ≥65 vs. <65 years occurred in 51% vs. 35% and 25% vs. 21% of patients in ERIVANCE BCC and EAS, respectively. To this regard, AEs leading to treatment discontinuation occurred in 15% vs. 11% and 11% vs. 2% of patients aged ≥65 vs. <65 years in ERIVANCE BCC and EAS, respectively.

Interestingly, in the STEVIE trial, 51 patients developed SCC during treatment, and many of these were >75 years with previous anamnesis of SCC, Bowen’s disease, or actinic keratosis.

Moreover, this analysis reported that elderly patients experienced clinical benefit from vismodegib similar to that for younger patients. Among patients with laBCC, the best overall response rate (BORR) was 46.7% and 72.7% in patients with laBCC aged ≥65 and <65 years, respectively. In the EAS, the BORR was 45.8% and 46.9% in patients with laBCC aged ≥65 and <65 years, respectively. Regarding patients with mBCC, the BORR was 35.7% and 52.6%, and 33.3% and 28.6% in patients aged ≥65 and <65 years, in the ERIVANCE BCC and EAS, respectively.

In addition, the efficacy and tolerability of vismodegib in patients with multiple or single comorbidities has been poorly investigated. Only a few reports have been published regarding the safety profile of Hh inhibitor in this frail population affected by multiple comorbidities, such as cardiovascular disease, severe liver dysfunction, immunosuppressive diseases, or hepatic chronic infection. Maul and colleagues published the first single case study reporting a good efficacy and tolerability of vismodegib in a patient with multiple comorbidities, such as chronic stage 4 kidney disease receiving dialysis treatment [66].

Finally, a recent exploratory retrospective analysis of eight elderly BCC patients with multiple comorbidities was conducted [67]. This retrospective study reported high tolerability and optimal safety profile of Hh inhibitor in elderly patients with severe multiple comorbidities. The safety profile of the study is in agreement with that derived from long-term follow-up of pivotal clinical trials, suggesting that Hh inhibitors represent a useful tool for the management of BCC patients affected by several comorbidities.

## 6. Predictive Biomarkers of Response to Hedgehog Inhibitors

Notwithstanding the exceptional clinical efficacy and long-term benefit of Hh inhibitors, there is no evidence for a preliminary best selection of patients defined as responders to targeted therapy. Although there has been extensive research in this field, the mechanisms of primary and adaptive resistance to Hh inhibitors has not been elucidated.

To date, several clinical characteristics have been reported as potential predictive markers of response to Hh inhibitors. Indeed, clinical features, such as young age [54], patients with laBCC, particularly with Gorlin’s syndrome, or advanced BCC patients who were without prior systemic exposure to Hh inhibitors or chemotherapy [56], have been reported to respond better to treatment.

In addition to the limited clinical characteristics, an option for exploring the behavior of BCC, includes the investigation of immunohistochemical (IHC) biomarkers correlated with different response rates, such as complete response, partial response, stable disease, or progressive disease, may make it easier to better select patients with advanced BCC who are most likely to respond to Hh inhibitors, and thus limit the use of an ineffective and useless therapy, in terms of adverse events and costs.

Indeed, IHC techniques could be easily employed, in a timely fashion, to determine a histological diagnosis of BCC, with limited costs, and this approach would appear to be particularly well suited to testing for a response to vismodegib.

In this context, an interesting study investigated whether a set of IHC markers, related to the tumor proliferation and migration ability, were associated with alternative pathways involved in basal cell proliferation or dissemination. The markers analyzed were CD56, platelet-derived growth factor receptor alpha (PDGF-Rα), CD117, matrix metalloproteinase 9 (MMP9), tissue inhibitor of matrix metalloprotease 3 (TIMP3), and CXC receptor 4 (CXCR4) [68].

The primary objective of this study was to describe the expression of CD56, PDGF-Rα, CD117, MMP9, TIMP3, and CXCR4 in advanced BCC. The secondary objective was to explore whether the expression of these markers was associated with a lack of response to vismodegib treatment in advanced BCC. This study reported that CD56 expression was significantly associated with risk of lack of response to vismodegib in advanced BCC cohort (OR = 5.5; 95% CI: 3.4–29.8; *p* = 0.0488), and a similar trend was observed for CXCR4 (OR = 3.45; 95% CI: 0.923–12.96; *p* = 0.066). However, the study is limited by the size of the population analyzed, due to the rarity of advanced BCC. Therefore, further studies will be needed.

Recently, a study investigated the expression levels of Hh pathway sixteen genes in a small sample size of laBCC as potential predictors of response. This study is the first to identify *GAS1* baseline levels, a critical Hh pathway gene, as a promising marker to predict the tumor response to vismodegib [69]. The authors speculate that the overexpression of *GAS1* strengthens the pathway enough to overcome vismodegib inhibition by releasing more SMO from PTCH1.

Interestingly, immune checkpoints inhibitors (ICI)s, such as anti-PD-1 monoclonal antibodies with or without Hh inhibitor, are currently being investigated in patients with advanced BCC. Unfortunately, the clinical benefit of ICIs is limited to a small group of treated patients. To this regard, the identification of predictive biomarkers of responsiveness to ICIs is one of the major challenges in cancer research [70]. Besides PD-L1 expression, other potential predictive biomarkers for ICIs have been investigated or are currently under evaluation also in advanced BCC. Interestingly, BCCs are reported to carry a high tumor mutational burden (TMB) (65 mutations/Mb), most likely due to the UV signature such as melanoma [71]. In an interesting study, Goodman et al. showed that the median TMB for 9 BCC samples and 1637 samples from other different type of tumors was 90/Mb and 4/Mb, respectively [72]. In addition, Ikeda et al. showed in two cases of BCC the amplification of the 9p24.3–9p22.2 region which contains the PD-L1, PD-L2, and JAK2 genes and typically predict the responsiveness to immunotherapy with anti-PD-1 mAbs in Hodgkin lymphoma [73]. Noteworthy, three out of four BCC patients treated with ICIs showed an objective and durable tumor response [74].

Unfortunately, we have no data regarding predictive biomarkers of response to Hh inhibitors in the elderly population affected by advanced BCC.

## 7. Case Reports

### 7.1. Case Report 1

This patient is a 93-year-old woman who presented a multifocal and ulcerated laBCC involving the whole nose, mostly to the left, and the frontal region, which recurred after prior surgical excision performed in 2015 (Figure 2a). The patient had a personal history of dementia hypertensive heart disease, type II diabetes, hyperuricemia, dyslipidemia, and osteoporosis in pharmacological therapy.

She had a performance status (PS) of 2 (Eastern Cooperative Oncology Group Performance Status (ECOG PS) = 2). Basal MRI demonstrated an enhancing lesion of the left nasal alar cartilage, extending posteriorly to the alar nasalis muscle (Figure 2f).

Following multidisciplinary discussion, in February 2017 the patient started treatment with Hh inhibitor, vismodegib at the recommended dosage of 150 mg daily. After 3 months of vismodegib treatment, she achieved a marked clinical partial response (Figure 2c). The clinical benefit was also correlated with radiological response (see Figure 2g,h).

Overall, vismodegib has been well tolerated. Specifically, treatment side effects included alopecia (grade 2), dysgeusia (grade 2), muscle cramping (grade 1), nausea (grade 2), decreased weight (grade 1), decreased appetite (grade 2), and fatigue (grade 2). In addition, cardiological evaluations, including electrocardiography and echocardiogram, remained unchanged during the course of Hh inhibitor, despite a history of heart disease.

Interestingly, following 12 months of vismodegib therapy, an erythematous lesion evolved in an ulcerated cutaneous lesion localized on the right cheek, requiring a radical surgical excision. The lesion was diagnosed as an SCC.

Following 39 months of vismodegib, the patient continues to remain in complete response (Figure 2e,h). In consideration of the clinical benefit and the good tolerance, the patient continues vismodegib treatment.

### 7.2. Case Report 2

This patient is a 92-year-old woman who presented an ulcerated laBCC involving the whole nose, mostly to the left and the inner canthus of the ipsilateral eye. Previously, the lesion had been treated with radical surgical excisions followed by local recurrence and rapid enlargement.

In anamnesis she reported hypertension, depression, previous ischemic stroke resulting aphasia and right hemiparesis, and was routinely treated with antithrombotic, antiepileptic, and antidepressant drugs, such as selective serotonin reuptake inhibitor.

She had a PS of 3 (ECOG PS = 3). Therefore, radiotherapy and surgical excision were contraindicated because of her important comorbidities and poor compliance.

CT scans of maxillofacial and orbits were performed at basal and showed neoplastic tissue of the inner canthus of the left eye with infiltration of the bone plane, the internal orbital vestibule, and the medial rectus muscle.

In March 2018, the patient started vismodegib therapy at the dosage of 150 mg daily (Figure 3a). Following 3 months, she achieved a partial response (Figure 3b) and then 6 months a complete clinical response (Figure 3c), which still persists after 27 months of Hh inhibitor (Figure 3e).

During the course of vismodegib treatment, the patient experienced minimal toxicity, including alopecia (grade 2), dysgeusia (grade 1), and muscle spasm (grade 2). Due to the relevant clinical benefit, the good safety profile and the absence of aggravation of the pre-existing comorbidities, vismodegib is still ongoing. Written informed consent was obtained from patients for publication of these case reports and any accompanying images.

## 8. Conclusions

The induction of specific molecular pathways, such as Hedgehog signaling cascade mediated by direct genetic modification in tumor cells, or paracrine signaling in the TME, plays an emerging hallmark in several human cancers, including BCC [75,76,77,78]. Therefore, SMO inhibitors, such as vismodegib and sonidegib, have allowed a revolution in the treatment of advanced BCC, thus, offering a successful therapeutic option for disfiguring and life-threatening disease associated with a long-term clinical benefit.

In consideration of the continuous increased aging of the general population and the contextual incidence increase of BCC, it is becoming increasingly urgent to establish the real advantage of treating elderly patients with Hh inhibitors.

To date, Hh inhibitor drugs should be used with caution in the elderly population. Indeed, the pivotal clinical trials did not include a sufficient number of patients older than 65; thus, it is unclear whether Hh inhibitors are safe to administer in a geriatric population. In addition, the BCC management in the ultra-aged population is commonly complicated by the concomitant presence of multiple comorbidities and poor compliance, which often contraindicate the conventional loco-regional therapies use.

In this regard, few and interesting data reported a similar clinical benefit from Hh inhibitors in the elderly compared to the young population. The safety profile of vismodegib in elderly patients appeared to be similar to that observed in the younger population. Moreover, emerging retrospective data and few clinical case reports show that advanced BCC treatment with Hh inhibitors among elderly patients with limited life expectancy and multiple comorbidities seem to be safe and effective in the clinical practice.

Therefore, Hh inhibitors can be used for the treatment of elderly patients with advanced BCC. Overall, the safety profile of vismodegib in elderly patients appears to be similar to that observed in younger populations. However, comorbidities or poor compliance of this neglected population must be considered when using Hh inhibitors.

Finally, recent advances in the understanding of the Hh signaling in both BCC progression and dynamic modulation of TME could suggest the potential synergism of the Hh inhibitors with immunotherapy [79]. In the near future, the use of immune checkpoint inhibitors could represent a valid and safe option even for elderly patients with advanced BCC who progress or are intolerant to prior Hh inhibitor therapy.

**Authors Contributions:** All authors (A.P., G.G., M.A., T.F., A.V., R.C.) listed have made a substantial, direct and intellectual contribution to the work, and approved it for publication. All authors have read and agreed to the published version of the manuscript.

## Figures and Tables

**Figure 1 ijms-21-08596-f001:**
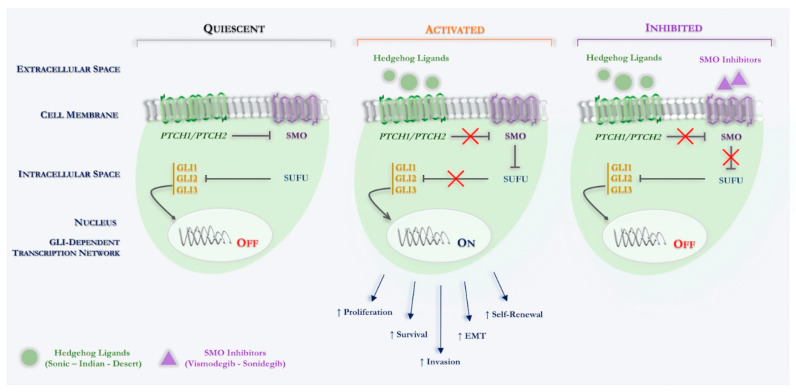
Schematic overview of canonical Hedgehog signaling in Basal Cell Carcinoma. Hh signaling pathway: quiescent: in the absence of an Hh ligand, PTCH inhibits SMO, thus, indirectly permitting SUFU to bind to, and thereby inactivate GLI transcription factors. Activated: binding of an Hh ligand to PTCH relieves PTCH-mediated inhibition of SMO, thereby inhibiting SUFU binding to GLI transcription factors, enabling the latter to enter the nucleus and activate Hh target genes. Inhibited: binding of a SMO inhibitor (vismodegib; sonidegib) relieves SMO-mediated inhibition of SUFU binding to GLI transcription factors, which are thereby inactivated. Abbreviations: PTCH, Patched receptor; Hh, Hedgehog; SMO, Smoothened; GLI, Glioma-associated oncogene homologue; SUFU, Suppressor of fused; EMT, Epithelial-mesenchymal transition.

**Figure 2 ijms-21-08596-f002:**
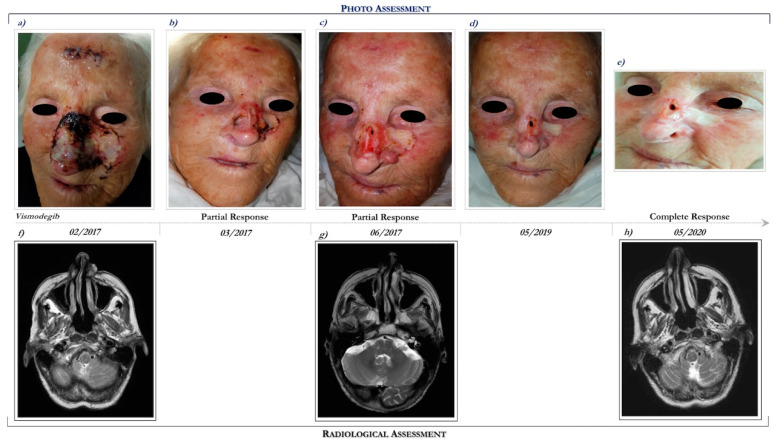
Clinical case report 1 of locally advanced BCC. Case report 1. Photo assessment: before vismodegib treatment (**a**); after 28 days of vismodegib (**b**); partial clinical response at 6 months (**c**); onset of complete clinical response (**d**); ongoing clinical benefit (**e**). Radiological assessment: basal cranial MRI performed in February 2017 before starting vismodegib (**f**), in June 2017 during vismodegib therapy (**g**) and in May 2020 following 27 months of vismodegib treatment (**h**).

**Figure 3 ijms-21-08596-f003:**
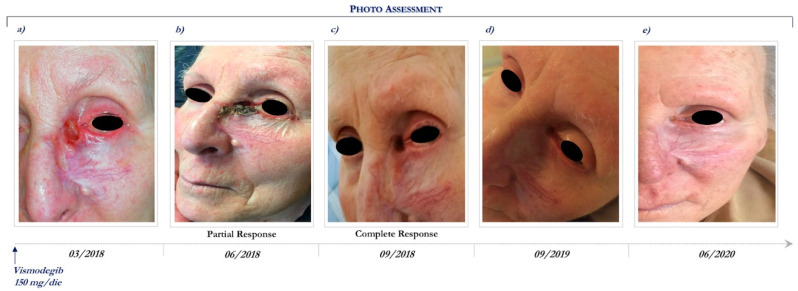
Clinical case report 2 of locally advanced BCC. Case report 2. Photo assessment: before vismodegib treatment (**a**); after 28 days of vismodegib (**b**); partial clinical response at 6 months (**c**); onset of complete clinical response (**d**); ongoing clinical benefit (**e**).

**Table 1 ijms-21-08596-t001:** Efficacy of Vismodegib in ERIVANCE trial (investigator review).

Results	Primary Analysis (Follow up 9 Months)		Long Term Analysis (Follow up 39 Months)	
	mBCC (*n* = 33)	laBCC (*n* = 63)	mBCC (*n* = 33)	laBCC (*n* = 63)
ORR n (%) (95%CI)	15 (45.5) (28.1–62.2)	38(60.3) (47.2–71.7)	16(48.5) (30.8–66.2)	38(60.3) (47.2–71.7)
CR	0	20	0	20
PR	15	18	16	18
SD	15	15	14	15
PD	2	6	2	6
Median DOR, m (95%CI)	12.9 (5.6–12.9)	7.6 (7.4–NE)	14.8 (5.6–17.0)	26.2 (9.0–37.6)
Median PFS, m (95%CI]	9.2 (7.4–NE]	11.3 (9.5–16.8]	9.3 (7.4–16.6)	12.9 (10.2–28.0)
Median OS, m(95%CI)	NE (13.9–NE)	NE (17.6–NE)	33.4 (18.1–NE)	NE (NE)
1 year OS %	75.5 (57.3–93.6)	91.6 (83.5–99.7)	78.7 (64.7–92.7)	93.2 (86.8–99.6)
2 year OS %	NE	NE	62.3 (45.4–79.3)	85.5 (76.1–94.8)

Abbreviations: mBCC, metastatic basal cell carcinoma; laBCC, locally advanced basal cell carcinoma; ORR, objective response rate; CI, confidence interval; CR, complete response; PR, partial response; SD, stable disease; PD, progressive disease; DOR, duration of response; PFS, progression-free survival; OS, overall survival; NE, not estimable.

**Table 2 ijms-21-08596-t002:** Efficacy of Vismodegib in STEVIE Trial.

Results	Locally Advanced BCC	Metastatic BCC	Total
Response rate *n* (%) [95%CI]	738 (68.5) [65.66–71.29]	31 (36.9) [26.63–48.13]	769 (66.2) [63.43–68.96]
Complete response, *n* (%)	360 (33.4)	4 (4.8)	364 (31.4)
Partial response, *n* (%)	378 (35.1)	27 (32.1)	405 (34.9)
Stable disease, *n* (%)	270 (25.1)	39 (46.4)	309 (26.6)
Progressive disease, *n* (%)	21 (1.9)	9 (10.7)	30 (2.6)
Missing or not evaluable, *n* (%)	48 (4.5)	5 (6.0)	53 (4.6)
Median time to response, *n*	1077	84	1161
months (95% CI)	3.7 (2.9–3.7)	NE (5.5-NE)	3.7 (3.5–3.7)
Median duration of response, *n*	738	31	175
months (95% CI)	23.0 (20.4–26.7)	13.9 (9.2-NE)	22.7 (20.3–24.8)
Median progression-free survival, *n*	1103	89	1192
months (95% CI)	23.2 (21.4–26.0)	13.1 (12–0-17.7)	22.1 (20.3–24.7)

Abbreviations: BCC, basal cell carcinoma; ORR, objective response rate; CI, confidence interval; CR, complete response; PR, partial response; SD, stable disease; PD, progressive disease; DOR, duration of response; PFS, progression-free survival; OS, overall survival; NE, not estimable.

**(a) ijms-21-08596-a:** 

	ERIVANCE BCC (N = 104)	EAS (N = 119)
	laBCC	mBCC	laBCC	mBCC
	≥65 years	<65 years	≥65 years	<65 years	≥65 years	<65 years	≥65 years	<65 years
	(*n* = 33)	(*n* = 38)	(*n* = 14)	(*n* = 19)	(*n* = 27)	(*n* = 35)	(*n* = 26)	(*n* = 31)
Median age, years	75	50.5	71.5	53	77	53	71.5	55
ECOG PS, *n* (%)								
0	22 (66.7)	29 (76.3)	5 (53.7)	8 (42.1)	12 (44.4)	27 (77.1)	14 (53.8)	16 (51.6)
1	7 (21.2)	8 (21.1)	9 (64.3)	10 (52.6)	12 (44.4)	7 (20.0)	10 (38.5)	12 (38.7)
2	4 (12.1)	1 (2.6)	0	1 (5.3)	3 (11.1)	1 (2.9)	2 (7.7)	3 (9.7)
Prior treatment, *n* (%)								
Surgery	27 (81.8)	35 (92.1)	14 (100)	18 (94.7)	25 (92.6)	32 (91.4)	25 (96.2)	29 (93.5)
Radiotherapy	13 (39.4)	9 (23.7)	9 (64.3)	10 (52.6)	11 (40.7)	9 (25.7)	18 (69.2)	17 (54.8)
Systemic therapy	2 (6.1)	6 (15.8)	2 (14.3)	8 (42.1)	7 (25.9)	4 (11.4)	7 (26.9)	13 (41.9)
Surgery contraindicated, *n* (%)	14 (42.4)	29 (76.3)	NA	NA	18 (66.7)	17 (48.6)	NA	NA

Abbreviations: BCC, basal cell carcinoma; laBCC, locally advanced BCC; mBCC, metastatic BCC; ECOG PS, Eastern Cooperative Oncology Group Performance Status; NA, not applicable.

**(b) ijms-21-08596-b:** 

	ERIVANCE BCC (N = 96)	EAS (N = 95)
	laBCC	mBCC	laBCC	mBCC
	≥65 years	<65 years	≥65 years	<65 years	≥65 years	<65 years	≥65 years	<65 years
	(*n* = 30)	(*n* = 33)	(*n* = 14)	(*n* = 19)	(*n* = 24)	(*n* = 32)	(*n* = 18)	(*n* = 21)
BORR, n (%)	14 (46.7)	24 (72.7)	5 (35.7)	10 (52.6)	11 (45.8)	15 (46.9)	6 (33.3)	6 (28.6)
Complete response	8 (26)	12 (36)	0	0	2 (8)	4 (12)	1 (6)	1 (5)
Partial response	6 (20)	12 (36)	5 (36)	10 (53)	9 (38)	11 (34)	5 (28)	5 (24)
Stable disease	11 (37)	4 (12)	7 (50)	8 (42)	12 (50)	15 (47)	9 (50)	11 (52)
Progressive disease	2 (7)	4 (12)	1 (7)	1 (5)	0	0	1 (6)	2 (10)
Not evaluable/missing	3 (10)	1 (3)	1 (7)	0	1 (4)	2 (6)	2 (11)	2 (10)

Abbreviations: BCC, basal cell carcinoma; laBCC, locally advanced BCC; mBCC, metastatic BCC; BORR, best overall response rate.

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
