# Peer review of "A Vismodegib Experience in Elderly Patients with Basal Cell Carcinoma: Case Reports and Review of the Literature"

_ijms, 2020, doi:10.3390/ijms21228596_

Round 1
Reviewer 1 Report
This is a very extensive and impressive review of the literature on hedgehog pathway inhibitors in the elderly population, in addition to two case reports.
However, I think the manuscript still needs much work. The English language requires major editing. Some of the data is inaccurate and some of the numbers cited do not match the values that appear in the cited sources. The manuscript also contradicts itself several times.
Author Response
Answers to the Reviewers’ comments:
We thank the Reviewer for general positive comments and suggestions.
As required, we revised the manuscript globally. Specifically, we reviewed the data and numbers cited in the text, exactly in Chapter 2 “Local treatment options for BCC”. In addition, we would like to let you know that the Review has been revised by a native English speaker.
Please see the attachment.

Reviewer 2 Report
The authors of the manuscript “A Vismodegib experience in elderly patients with basal cell carcinoma:case reports and review of the literature” wrote a review about the role of HH signaling in BCC pathogenesis and what this implicates for elderly patients with respect to treatment options. In addition, they provide two case reports of two elderly women, who received vismodegib because of advanced BCC.
First, the authors discuss local treatment options that are currently available in the clinics. It would be great if the authors could make a short summary with regard to elderly patients. For me it was not clear, which of the local treatments is the best for the elderly. In addition, in line 80 - 83 the authors explain that wound healing after surgical procedures in geriatric patients is sometimes not straight forward, whereas in line 98 they wrote that standard surgery is well tolerated by elderly patients.
Second, the authors summarize the current knowledge of the role of Hedgehog signaling in BCC. However, they clearly should state that the most common mutations in BCC are PTCH1 mutations, whereas it is not clear whether PTCH2 mutations play a role in BCC (see e.g. Altaraihi et al. Human Genome Variation (2019) 6:10 https://doi.org/10.1038/s41439-019-0041-2). In addition, in Figure 1, it seems as if SMO is directly blocking the action of SUFU. However, since this is not the case (SMO interacts with Costal2), the authors should avoid showing such an interaction. In addition, it would be better to indicate “PTCH” instead “PTCH1/PTCH2” (please see my comment above). Furthermore, the authors should indicate that the overview of the HH pathways is extremely simplified and that it shows the major components that are either involved (e.g. GLI) or mutated (PTCH1 inactivating mutations, SMO activating mutations and SUFU inactivating mutations) in BCC (please see e.g. Bakshi et al, Mol Carcinog. 2017 December ; 56(12): 2543–2557. doi:10.1002/mc.22690).
I also did not understand the sentence “The genomic characterization has identified several mutations both downstream of Gli and independent of the Hh signaling that may be considered critical for the development or progression of BCC” (line 193/194). What exactly do the authors mean? Actually, the whole part from line 192 – 204 is somehow confusing. For example, HH inhibitors do not inhibit proliferation of BCC if the mutation is downstream of SMO. It would be great if the authors could comment on that: Thus, so far only the HH inhibitors, which are in fact SMO inhibitors, Vismo and Sonidegib have been approved for BCC (at least in Europe). However, these inhibitors only work if the mutation is upstream of SMO. That means they do not work if the BCC has a SMO or SUFU mutation.
In the next part (“4. Hedgehog pathway inhibitors”) the authors summarize the findings of the big trials that have used Vismodegib or Sonidegib for treatment of metastatic or advanced BCC. In this section they also can report the findings that vismodegib and sonidegib can promote immune responses in Basal Cell Carcinoma.
Then I would move the current part 6 “Safety and efficacy of Vismodegib in advanced BCC elderly patients” (please note that the word “of” is missing) in position 5 and would omit the current section 5 “Predictive biomarkers of response to Hedgehog inhibitors”. Since the focus of the review should be on elderly patients and HH signaling, I did not get the point of this section, especially since there are no biomarkers today that may predict the response of BCC towards SMO inhibitors. In this section, the authors rather discuss biomarkers that may predict the response of BCC towards immune checkpoint inhibitors.
Finally, in the conclusion section I do not understand why the authors have included the citations 74 and 75 (about EGFR and CD73). In addition, it would be great if the authors could bring this section to the point: As far as I understood and based on the currently available data, HH inhibitors can be used for treatment of the elderly. Indeed, the safety profile of vismodegib in elderly patients appears to be similar to that observed in younger population. However, comorbidities or poor compliance of the patients must be considered when using SMO inhibitors.
Author Response
Answers to the Reviewers’ comments:
We thank the Reviewer for general positive comments and suggestions.
In agreement to the suggestions, we have addressed point-to-point the minor comments of the Reviewer.
- Reviewer asks to clarify and summarize which of the local treatments is the best for the elderly patient affected by BCCs.
We thank the Reviewer for this interesting suggestion. Unfortunately, to date, no clinical data are available concerning the optimal management of elderly patients affected by BCC. There is no one-size-fits-all recommendation for the care of these tumor. However, according to your suggestion, we have tried to report and summarize the main clinical data deriving from retrospective analyzes regarding the effectiveness, benefits and risks of common surgical and nonsurgical treatments for the elderly patient with BCCs (see chapter “Local treatment options for BCC”, page 2-4). An appropriate reference has been included (#12: National Comprehensive Cancer Network. NCCN Clinical Practice Guidelines in Oncology: Basal Cell Skin Cancer, Version I. 2019 Published August 31, 2018).
- Reviewer suggests to better define the role of PTCH1/2 genes in the BCC pathogenesis and modify Figure 1.
As reported in the manuscript, we clearly reported the frequency of PTCH1 gene mutation considered critical for the development of BCC (see page 5, line 209). As suggested by Reviewer, we now specified the role of PTCH2 gene mutation in BCC pathogenesis (page 5, line 216), and added appropriate reference (#49: Altaraihi M, Wadt K, Ek J, Gerdes AM, Ostergaard E: A healthy individual with a homozygous PTCH2 frameshift variant: Are variants of PTCH2 associated with nevoid basal cell carcinoma syndrome? Hum Genome Var 22;6:10, 2019. In according to the suggestions, the Figure 1 is now modified (see Figure 1 and legend); and we specified that the Hh signalling pathway has been reported in a simple and concise way (line 189, 198, page 5).
- Reviewer asks to better specify the spectrum of action of SMO inhibitors (Vismodegib, Sonidegib) based on the types of mutations reported.
As required, we modified the unclear sentence. In addition, we now inserted a paragraph in order to specify the exact spectrum of action of SMO inhibitors based of type mutations (page 6, line 223-229).
- Reviewer suggests to describe in the section 4 ‘Hedgehog pathway inhibitors’ the immune modulation of tumor microenvironment induced by vismodegib and sonidegib.
As required, we added the novel findings regarding the potential immune modulation of tumor microenvironment mediated by vismodegib and sonidegib (line 324-330, page 8).
- Reviewer suggests to move the current section 6 “Safety and efficacy of Vismodegib in advanced BCC elderly patients” in position 5 and proposes to omit the section 5 “Predictive biomarkers of response to Hedgehog inhibitors”.
As suggested, we moved the section 6 in position 5.
Regarding to the Reviewer’s request to remove the “Predictive biomarkers of response to Hedgehog inhibitors” section from the manuscript, we believe it is appropriate to keep it in the text for two reasons. First, in the era of precision medicine and in the context of the topic addressed, it is necessary to report the data, albeit disappointing, of the search for potential predictive biomarkers of response to Hh inhibitors. This would be even more relevant when referring to a ‘fragile’ population such as the elderly, where the use of biomarker in clinical practice could be of great help in deciding whether or not treat these patients. Second, in consideration of the promising albeit preliminary results of the use of immunotherapy for BCCs treatment, we believe that the data currently available regarding the predictive biomarkers to response to immunotherapy can be useful especially for the future selection of elderly patients affecting by BCCs. Interestingly, this strategy treatment in other cancer types is considered safe also in elderly patients.
- Reviewer asks to add a paragraph in the ‘Conclusions’ section that contains the conclusive message of the manuscript.
According to the appropriate suggestion, we insert in the Conclusion section the concise paragraph proposed reporting the final message of the manuscript (page 13, line 509-512). Furthermore, the 75-78 References have been inserted in order to report, as written in the sentence “…The induction of specific molecular pathways such as Hedgehog signalling cascade mediated….an emerging hallmark in several human cancers…”, how the induction of other molecular intracellular pathways (for example mediated by both BRAF in melanoma and EGFR in NSCLC) are able to shape the tumor microenvironment.

Reviewer 3 Report
This review article reports the role of Hedgehog signaling pathway in pathogenesis of cutaneous basal cell carcinoma (BCC) and treatment of BCC with Hedgehog signaling inhibitors. The authors also reported two case reports of BCC elderly patients treated with Hedgehog inhibitors. The manuscript is well organized and comprehensively written. I have a few minor comments.
- Line 28. “…focusing on the Hedgehog pathway signaling…”, is it more common to use “signaling pathway”?
- Line 161. “The skin contains several stem cells that…”, I suggest changing to “several types of stem cells” or “different stem cells”.
- Line 257. For the description of vismodegib effects in pregnant rat model, literature reference is needed.
Author Response
Answers to the Reviewers’ comments:
We thank the Reviewer for general comments and suggestions.
In agreement to the suggestions, we have addressed point-to-point the minor comments of the Reviewer.
- Reviewer asks to modify the sentence “Hedgehog pathway signalling” in “Hedgehog signalling pathway”.
According to the suggestion, we re-wrote the sentence throughout the text (page 1, line 28; page 4, line 174; page 5, line 190; page 6, line 223).
- Reviewer suggests to change the sentence “The skin contains several stem cells that…” I suggest changing to “several types of stem cells” or “different stem cells”.
As suggested, we have now modified the sentence (page 4, line 175).
- Reviewer asks to report a reference regarding the description of vismodegib effects in pregnant rat model.
We have now included appropriate sentence in the text (#56: Morinello E, Pignatello M, Villabruna L, Goelzer P, Burgin H: Embryofetal development study of vismodegib, a hedgehog pathway inhibitor, in rats. Birth Defects Res B Dev Reprod Toxicol 101:135-43, 2014).

Round 2
Reviewer 2 Report
In my opinion the review now reads well.
However, please insert "of" in the heading "Safety and efficacy of Vismodegib in advanced BCC elderly patients", which should read "Safety and efficacy of Vismodegib in advanced BCC of elderly patients"